# Added Value of Ultrasound-Based Multimodal Imaging to Diagnose Hepatic Sclerosed Hemangioma before Biopsy and Resection

**DOI:** 10.3390/diagnostics12112818

**Published:** 2022-11-16

**Authors:** Feiqian Wang, Kazushi Numata, Hiromi Nihonmatsu, Makoto Chuma, Naomi Ideno, Akito Nozaki, Katsuaki Ogushi, Mikiko Tanab, Masahiro Okada, Wen Luo, Masayuki Nakano, Masako Otani, Yoshiaki Inayama, Shin Maeda

**Affiliations:** 1Ultrasound Department, The First Affiliated Hospital of Xi’an Jiaotong University, No. 277 West Yanta Road, Xi’an 710061, China; 2Gastroenterological Center, Yokohama City University Medical Center, 4-57 Urafune-cho, Minami-ku, Yokohama 232-0024, Japan; 3Division of Diagnostic Pathology, Yokohama City University Medical Center, 4-57 Urafune-cho, Minami-ku, Yokohama 232-0024, Japan; 4Department of Radiology, Nihon University School of Medicine, 30-1 Oyaguchi Kami-cho, Itabashi-ku, Tokyo 173-8610, Japan; 5Department of Ultrasound, Xijing Hospital, Air Force Military Medical University, No. 127 Changle West Road, Xi’an 710032, China; 6Tokyo Central Pathology Laboratory, 838-1, Utsukimachi, Hachioji 192-0024, Japan; 7Division of Gastroenterology, Yokohama City University Graduate School of Medicine, 3-9 Fukuura, Kanazawa-ku, Yokohama 236-0004, Japan

**Keywords:** hepatic sclerosed hemangioma, contrast-enhanced ultrasound, elastography, B-flow, multimodal image

## Abstract

Imaging methods have the overwhelming advantage of being non-invasive in the diagnosis of hepatic lesions and, thanks to technical developments in the field of ultrasound (US), radiation exposure can also be avoided in many clinical situations. In particular, contrast-enhanced US (CEUS) outperforms other radiological methods in regard to real-time images, repeatability, and prompt reporting and demonstrates relatively few contraindications and adverse reactions. In this study, we reported in detail a rare benign tumor: hepatic sclerosed hemangioma (HSH). We described US-based multimodal imaging (B-flow imaging, US elastography, and Sonazoid CEUS) features of this HSH case. Furthermore, by summarizing the recently published literature on the imaging diagnosis of HSH, we offered readers comprehensive knowledge of conventional imaging methods (CT, MRI) and CEUS in the diagnosis of HSH and preliminarily discussed their mechanism of pathology-based diagnosis. Our multimodal imaging approach may provide a diagnostic strategy for HSH, thus avoiding unnecessary biopsy or resection.

## 1. Introduction

Hepatic sclerosed hemangioma (HSH) is identified as a rare benign tumor of the liver. Because of its characteristic of insidious onset and unclear etiology, very little related information is available in the literature and inexperienced clinical doctors may easily misdiagnose HSH as malignancy [1], such as hepatic metastatic carcinoma [2,3], hepatocellular carcinoma (HCC), biliary cystadenocarcinoma [4], gallbladder carcinoma [5], or intrahepatic cholangiocarcinoma [6].

The emergence of any disease in humans, including HSH, is due to complex biological processes and the interaction of various pathogenic factors. Therefore, the efficiency of a single-imaging approach for diagnosis HSH is supposed to be insufficient. Some researchers recommend that the combined use of a variety of imaging modalities may be a wise choice for the noninvasive diagnosis of HSH [7]. In recent years, ultrasound (US)-derived novel-imaging approaches have been increasingly used in diagnosing hepatic lesions. Contrast-enhanced US (CEUS) has well-known advantages, including, but not limited to, being relatively inexpensive, easy to perform, conducive to instant interpretation, reproducible in real time, and not reliant on radiation exposure [8]. However, there are very few studies on newly developed US techniques for the diagnosis and follow-up of HSH [9].

We have herein comprehensively applied US-based multimodal imaging to one HSH patient to make a more accurate diagnosis. “US-based multimodal imaging” in this study refers specifically to many imaging techniques derived from US, such as newly developed CEUS, B-flow, and US elastography, as well as the well-known techniques as gray-scale US and color Doppler. Computed tomography (CT) and magnetic resonance imaging (MRI) are the currently conventionally applied imaging methods for the diagnosis of HSH. Thereafter, we performed a detailed review of the literature that use CT, MRI methods, and CEUS to diagnosis HSH. There has been no report on the malignant transformation of HSH to date. Therefore, making full use of US-based multimodal imaging and other imaging methods (CT/MRI) might avoid patients undergoing unnecessary biopsies, surgical resections, and other over-diagnoses and treatments. In short, this study aimed to explore the characteristics of available US-based multimodal-imaging technologies in order to improve the diagnosis of HSH.

## 2. Materials and Methods

### 2.1. Data Collection

On the one hand, we retrospectively collected all the data of one pathologically confirmed HSH from our hospital. Written consent was obtained from this patient for publishing his information. Nevertheless, our report does not contain any personal information that could lead to the identification of the patient. All data collection and diagnostic and therapeutic procedures performed with this patient were in accordance with the principles of the Declaration of Helsinki.

On the other hand, CT/MRI and CEUS data for the diagnosis of HSH were obtained systematically from the literature. In detail, we searched the PubMed bibliographic database. We retrieved manually by using a search strategy of ((sclerosed hemangioma (Title/Abstract)) OR (sclerosing hemangioma (Title/Abstract))) AND ((hepatic (Title/Abstract)) OR (liver (Title/Abstract))). We found 35 studies describing CT/MRI and another 3 describing CEUS in PubMed from January 2010 to October 2022. All these cases made conclusive HSH diagnoses via pathology. In the consideration of the rapid development of imaging technology and radiological equipment, we did not include studies published before 2010. Thereafter, we excluded 13 studies, because (1) they were not direct cases but taken from other publications (*n* = 3); (2) the text was written in Japanese (*n* = 1); (3) they contained very little information on CT/MRI (*n* = 7); (4) several kinds of hepatic lesions were described together and the imaging features of HSH lesions cannot be extracted from their mixed data (*n* = 2). Finally, 22 published case–control studies, case series, and case reports written in the English language were enrolled in this study.

### 2.2. Data Analysis

Our case was described in detail, from the detection of the hepatic lesion to the diagnosis of HSH and every follow-up. Regarding the retrieved literature, the features of the 4 cases using CEUS (including our own case) and the 22 cases using CT/MRI for HSH diagnosis are respectively displayed in tables. Owing to the possible heterogeneity of cases collected from 22 different studies and the lack of a control group with homogeneous baseline characteristics, no statistical analysis was performed.

## 3. Results

### 3.1. Our Case

A 57-year-old male patient was found to have a hepatic lesion during a physical examination via unenhanced CT examination. It revealed a diffuse hypo-intense area in segment V/VIII of the liver with ill-defined borders and an irregular shape (Figure 1A). The patient did not report any specific symptoms. No special disease history was reported, except allergic asthma (diagnosed 25 years ago). There were no positive findings of the physical examination. A thorough laboratory workup showed that routine blood, urine, feces, serum glucose level, tumor markers, electrolytes, liver function, and kidney function were all within normal ranges, except for a slight increase in carcinoembryonic antigen (8.7 ng/mL). Hepatitis B and C serology was negative. The electrocardiogram and chest X-ray were uneventful.

We carried out a series of US-related examinations on a LOGIQ E9 US system (GE Healthcare, Milwaukee, WI, USA). The lesion appeared as a slight hyperechoic area with an unclear boundary in grayscale US (Figure 1B). Color Doppler imaging exhibited very few vessels within the lesion (Figure 1C) and a relatively rich blood flow signal in the surroundings. B-flow imaging exhibited very faint blood flow with peritumoral strips of blood vessels (Figure 1D). US elastography imaging (independently performed by K.N. and H.N with more than ten years’ experience in liver US diagnosis) revealed dotted and flaky red–green mixed images (Figure 1E), which suggests a moderate to soft texture. Afterwards, CEUS, using Sonazoid^®^ contrast agent (GE Healthcare, Oslo, Norway),was performed according to a previous study [10]. Enhancement from the peripheral to the central area in the arterial phase (AP) was detected (Figure 1F). Slight hypovascularity and marked hypoechogenicity were exhibited, respectively, in the portal phase (PP) (Figure 1G) and the post-vascular phase (PVP) (Figure 1H). The lesion was observed showing hypervascularity when the contrast agent was re-injected in PVP (Figure 1I).

A percutaneous needle biopsy under the guidance of SCEUS was performed. Impressively, a proliferation of elastic fiber, a high density of hyalinized collagenous tissue, and extensive fibrogenesis were found (Figure 2A–D). Within the area of fibrogenesis, hepatocytes and the bile duct had basically disappeared (Figure 2E). Around the area of fibrogenesis, blood vessels, especially arterioles, proliferated (Figure 2F,G). Notably, abundant mast cells, which are considered a pathological feature for the deterministic diagnosis of HSH [11], were detected by CD117 staining (Figure 2H,I). After consensus, two experienced pathologists (N.T. and M.N., working in the field of liver pathology for 20 years) diagnosed the lesion as an HSH.

The patient was left untreated and was closely followed-up after discharge. In the following two years, the size of the lesion gradually decreased in the US images. The size was 32 mm, 27 mm, 25 mm, 20 mm, 19 mm, and 18 mm at the time of initial detection, and at 2, 3 (before biopsy), 11, 20 and 36 months after detection, respectively. Consistently, the size decreased in the 36th month unenhanced MRI (Figure 3N). Within three months of detection, the lesion showed hypervascularity in AP (Figure 3A–C), whereas from the 11th month after the detection, the lesion was found to be hypovascularized in AP (Figure 3D–F). Throughout the whole process of follow-up, the lesion exhibited a consistent washed-out appearance in PVP, similar to that in the first detection (Figure 3G–L). As time went on, the intensity of the contrast agent perfusion was reduced. In particular, it showed very faint perfusion in the most recent follow-up of the 36th month. Notably, the elastography image of the latest follow-up revealed red-dominated images (Figure 3P), which suggested a soft texture.

### 3.2. The Characteristic Findings of CEUS in the Diagnosis of HSH

Four cases using CEUS diagnosis of HSH (including our case) were reported by East Asia scholars. They all reported a single lesion. Three of them presented unclear margins under US examination. Regarding common features of CEUS appearance, they exhibited wash-in in AP and wash-out in PVP (Table 1). The dynamic observation of the contrast agent perfusion showed that all these lesions underwent peripheral enhancement.

### 3.3. The Characteristic Findings of CT/MRI in the Diagnosis of HSH

Table 2 shows that, of these enrolled 22 studies, there were 76 HSH lesions from 73 patients. Most (17/22) studies were from research institutes and/or hospitals in East Asia. Epidemiologically, most of the patients were aged (median: 58.8, range 17–79 years old). There was no significant difference between the male and the female gender in the occurrence of HSH (38/35). Tumor markers, transaminases, and bilirubin values were almost within normal limits. Only a few (12.3%, 9/73) of the patients had cirrhosis. The lesions were mostly solitary (96.1%, 73/76). The majority (73.9%, 17/23) of the lesions were located in the right hepatic lobe.

When observed by CT/MRI (Table 3), the lesions of HSH were more likely to show an irregular or unclear margin (78.0%, 46/59). Most of the lesions (73.3%, 22/30) were heterogeneous. Calcification (6.6%, 5/76) and capsular retraction (10.5%, 8/76) were seldom present. Most HSH lesions showed arterial enhancement (90.8%, 69/76) in contrast-enhanced CT and/or MRI. When performing contrast-enhanced MRI, hyperintensity was commonly seen in the T2-weighted image (95.2%, 59/62), while hypointensity frequently appeared in both the T1-weighted (89.7%, 50/69) and the hepatobiliary phase image (92.1%, 41/44). The dynamic enhancement patterns of centripetal (39.5%, 30/76) and peripheral (55.3%, 42/76) were commonly detected.

## 4. Discussion

As seen in the cases we collected, there are very few valuable positive epidemiological and laboratory characteristics of HSH, which suggests that doctors may turn to radiological methods such as US, CT, and MRI for help.

Reviewing the literature on US, contrast-enhanced CT, and MRI, some hidden imaging features of HSH can be extracted. Interestingly, some of the features that appear to be prone to misdiagnosis can be reasonably explained in perspective of the pathological generation of HSH. The first feature is that most HSH lesions have unclear margins, which differs from the well-delineated characteristics of normal hemangiomas. Secondly, most HSH cases showed heterogeneity, regardless of lesion size. This might be caused by the variety of complex compositions within HSH, such as extensive hyalinization, the fibrogenesis of vascular space, and the proliferation of blood vessels [6], which are important microscopic features of HSH [1,11]. In contrast, a common hemangioma is generally homogeneous. Heterogeneity only presents when the hemangioma grows larger (more than 4 cm) and hyalinization develops [29]. The third feature is quick wash-in and wash-out after the contrast administration. This “trap” appearance (might cause a false diagnosis of malignancy) can be explained by the abundant arteries detected microscopically by ɑ-SMA staining. The hypoechoic area in the PVP of CEUS and the hypointensity in the hepatobiliary phase of EOB-MRI indicate a lack of normal functional hepatocytes to take up the contrast agent. Therefore, when HSH develops, there are few normal hepatocytes in the lesion. Fourth, when we further explored the detailed perfusion pattern of contrast-enhanced imaging, characteristic peripheral enhancement and centripetal filling were revealed for most HSH lesions. These appearances are generally considered to be typical features of hemangioma. Nevertheless, it is worth noting that the central unenhanced area was actually the sclerotic mesenchyme for HSH, not necrosis for normal hemangioma [27]. Fifth, almost all the HSH lesions were hypointense on T1-weighted MR images and hyperintense on T2-weighted MR images. The hyperintensity on T2-weighted imaging was thought to be attributable to slow blood flow in the vascular spaces [30] and the richness of hyalinization in the lesion [7], which can be respectively explained by the dilation of hepatic sinusoids and the extensive hyalinized degeneration of HSH. Finally, with the follow-up, the blood supply of the lesion decreased gradually. During the same time period, the size of the lesion decreased or even disappeared. Consistently with our case, Doyle et al. reported an HSH case with a size that decreased over time, with the reduction or even disappearance of regions previously seen by contrast-enhanced MRI [31]. We inferred that the small vessels in the lesion progressed from dilatation to stenosis, or even occlusion.

“US-based multimodal imaging” has many advantages. Because of its real-time nature, US-based multimodal imaging has a high temporal resolution. Meanwhile, high resolution can be obtained by using a high frequency probe. In recent years, newly developed US modality (such as Superb Microvascular Imaging) was believed to assess blood perfusion of lesions almost by a microvascular perspective. Unlike previous cases, we have revealed some valuable unreported features of “US-based multimodal imaging”. More impressively, the characteristics from US elastography and B-flow imaging of our HSH patient point to a correct diagnosis of benignity. The strain elastography technique measures tissue stiffness by applying external pressure to make a tissue dimensionally deform [32]. It is generally acknowledged that the hardness of a malignant tumor is much higher than that of the background liver at the same depth. The findings of US elastography suggest the lesion was benign. B-flow imaging neither reduces the frame rate nor suffers from artifacts such as blooming or wall overwriting, as the Color Doppler mode does; thus, this technique can display the true hemodynamics of the lesion in real time, especially at low speed [33,34]. B-flow imaging was reported to be used to distinguish hepatic lesions between benign and malignant ones. For example, hemangiomas have no (50%) or multiple vessels (40%) that are statistically different from the rich blood manifestation of the basket pattern (57.8%) of HCC and the penetrating vessels pattern (60.9%) of metastatic carcinoma [35]. The appearance of B-flow imaging in our case (faint blood flow) suggests a possible diagnosis of a benign mass.

The conclusions of the HSH literature, especially those of our own case, have enlightened us as to some critical implications for the diagnosis of HSH. Firstly, when contrast-enhanced CT/MRI imaging methods are unavailable, US-based multimodal imaging may be the preferred complementary modality. Taking our case and another HSH case [12] as an example, both patients had a medical history of asthma and, accordingly, they had a high risk of being allergic to gadolinium chelates. Therefore, contrast-enhanced CT or MRI examination were not performed. In contrast, it is absolutely safe for these types of patients to undergo CEUS, US elastography, and B-flow imaging, especially when radiological examinations are frequently used during follow-up. Secondly, for non-cirrhotic patients, US-based multimodal imaging can help doctors make better clinical decisions. For example, if doctors only focus on the single modal imaging CEUS, the “quick wash-in and wash-out” phenomenon would possibly lead to a wrong malignant diagnosis. In comparison, US-based multimodal imaging provides doctors with a wealth of diagnostic information. Both US elastography imaging and B-flow imaging suggest benign diagnoses. In this setting, doctors can safely follow up patients rather than aggressively adopt an invasive diagnosis and/or treatment. Last but not least, the appearances of the imaging findings are always closely related to the underlying biological behavior and histopathological changes of the disease. In terms of the radiological findings of HSH, which mimic primary or metastatic malignancy, we should concentrate on mining the relationship between the inherent pathology mechanism and the external imaging phenotype of HSH to better understand the imaging features of HSH.

The limitations of our study should be acknowledged. First, we only carried out descriptive analysis, rather than statistical analysis, on the imaging features of HSH. In order to obtain more reliable conclusions, the imaging features of HSH should be analyzed statistically in comparison with those of other liver malignancies. Second, there may be inevitable heterogeneity bias among the different studies included in this study. Third, though US-based multimodal imaging is promising in the diagnosis of HSH, we should keep in mind that they still have some insurmountable defects. In certain patients (cirrhosis or fatty liver) or lesions (deep-located, covered by ribs or lung gas, isoechogenicity), neither US nor US-based multimodal imaging can provide good visualization for an accurate diagnosis [36].

## 5. Conclusions

In conclusion, our study has preliminarily revealed some characteristics of HSH using US-based multimodal imaging, especially grounding a novel understanding of B-flow imaging and US elastography. Based on the characteristics of imaging techniques summarized in the recent literature and in our case, the non-invasive diagnosis of HSH is possible. However, large-sample data and statistical analyses based on HSH are still needed in the future if reliable conclusions are to be drawn for clinical diagnoses.

## Figures and Tables

**Figure 1 diagnostics-12-02818-f001:**
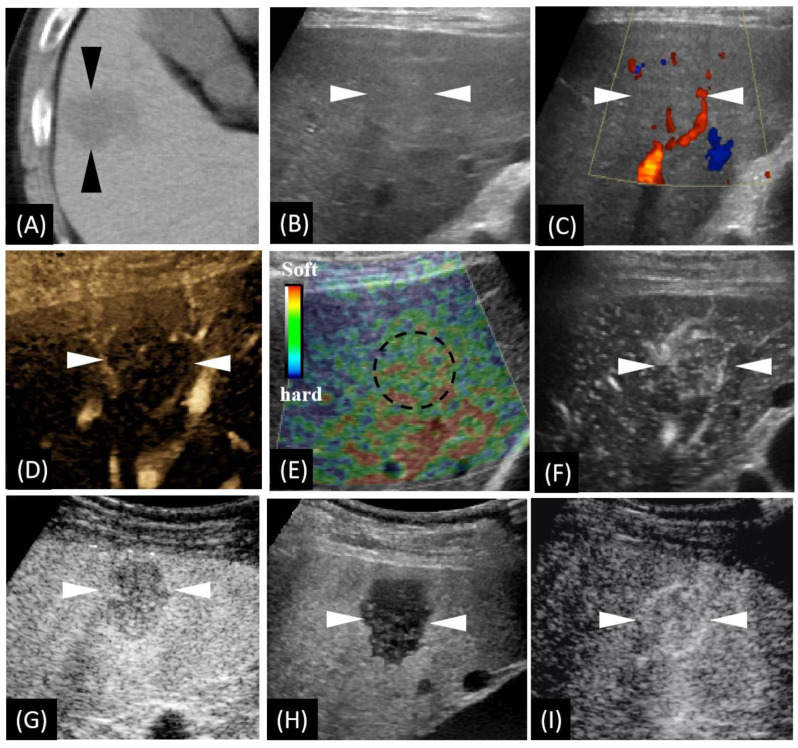
Unenhanced computed tomography (CT) and ultrasound (US)-based multimodal imaging of sclerosed hemangioma (HSH) at first detection (our case, a 57-year-old male patient). (**A**) Axial CT image shows the largest cross-sectional area of the tumor in the routine unenhanced phase. (**B**) Grayscale US shows a slightly hyperechoic lesion in segment V, near the surface of the liver. There is neither halo sign surrounding nor mosaic sign within the lesion. (**C**) Color Doppler flow imaging shows there is no significant blood flow within the lesion, while there is a relatively rich blood flow signal in the surrounding liver parenchyma. (**D**) B-flow imaging exhibited a central faint blood flow and peritumoral strips of blood vessels. (**E**) US elastography showed dotted and flaky red and green mixed images dominated by green. (**F**) Using low mechanical index (MI) harmonic imaging, arterial phase (10–50 s after initiation of injection) Sonazoid contrast-enhanced US shows hypervascularity with centripetal vessels. (**G**,**H**) Slight hypovascularity is shown in the portal phase (80–120 s after initiation of injection) (**G**), while marked hypoechogenicity with distinct margins is shown in the post-vascular phase (10 min after initiation of injection) (**H**). (**I**) Re-injection of Sonazoid agent shows hypervascularity in the arterial phase. The arrowheads seen in images **A**–**D** and **F**–**I** and the dotted circle in image **E** indicate the margin of the lesion.

**Figure 2 diagnostics-12-02818-f002:**
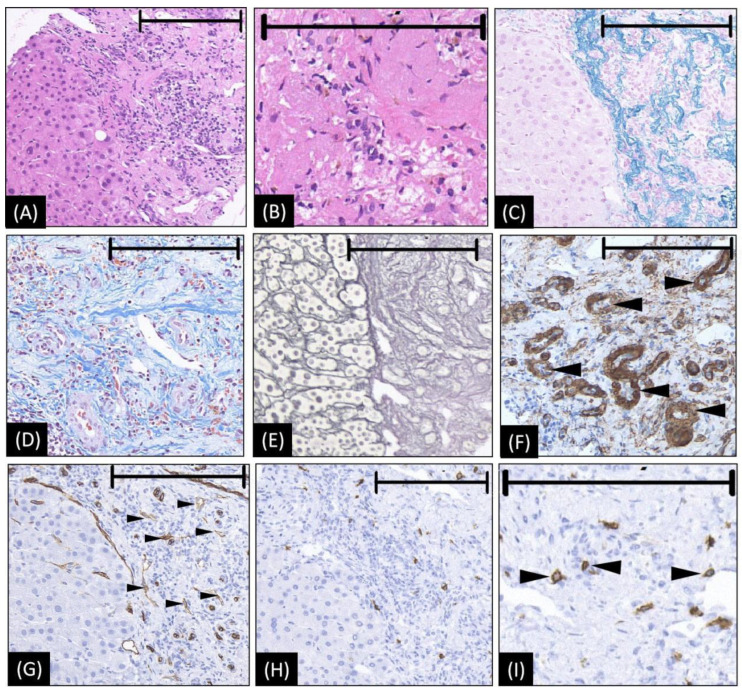
The microscopic appearance of biopsy tissue from HSH (our case). (**A**,**B**) Hematoxylin–eosin staining reveals abundant collagenous tissue around and between the small vessels. Within the area of fibrogenesis, the hepatocytes and bile duct basically disappeared. (**C**) Victoria blue staining shows abundant elastic fibers of varied thickness are dispersed in the stroma. (**D**) Masson trichrome staining showed a high density of hyalinized collagenous tissue stained in blue around and between small vessels. (**E**) The silver stain shows extensive fibrogenesis. Within the area of fibrogenesis, the hepatocytes and bile duct basically disappeared. The frame of the reticular fiber has changed to a great extent (**F**) The staining of actin on the arterial wall by ɑ-SMA, indicating an increase in arterioles (as arrowheads denoted). (**G**) CD34 is diffusely expressed in endothelial cells of the vessels (small black arrowheads), suggesting the blood vessels proliferated in the sclerosed portion. (**H**,**I**) CD117 immunohistochemical stains show increased mast cells (black arrowheads in (**I**)), which is considered strongly related to angiogenesis and connective tissue hyperplasia. In images **A**, **C**, **E**, **G**, and **H**, the left halves show the normal liver area as a reference, while the right halves are the lesion area to show the positive histological changes. The length of the black bar in the upper side of each figure represents 200 microns (µm) as a magnification reference.

**Figure 3 diagnostics-12-02818-f003:**
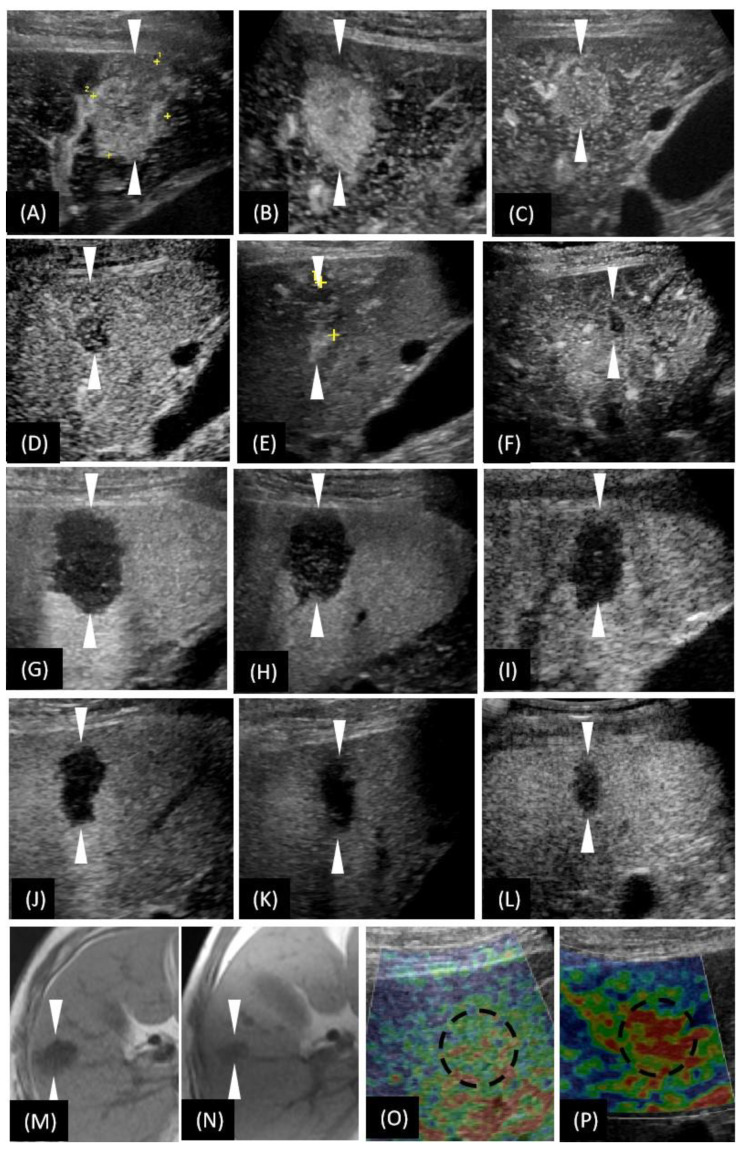
T1-weighted images of magnetic resonance imaging (MRI) and the US-based multimodal imaging appearance of HSH (our case) in each examination of follow-up. (**A**–**F**) A certain time point of AP; (**G**–**L**) PVP by CEUS examination. (**A**–**L**) are listed in the order of first detection, 2nd month, 3rd month (the day before biopsy), 11th month, 20th month, and 36th month, respectively. (**M**,**N**) unenhanced MRI; (**O**,**P**) US elastography. (**N**,**P**) were taken in the 36th month. To make a more intuitive comparative observation, we have included (**M**,**O**) here, whose images were taken at the time of the first detection of the lesion. Obviously, the size of the lesion was reduced (unenhanced MRI), the vascularity was reduced (CEUS), and the texture softened (US elastography). The arrowheads seen in images **A**–**H** and the dotted circle in image **O**–**P** indicate the margin of the lesion.

**Table 1 diagnostics-12-02818-t001:** CEUS characteristics of three patients with HSH ^1^.

Author, Year	Our Case (Wang F.)	Ando Y., 2018 [9]	Akahoshi S., 2020 [12]	Yoo J., 2020 [13]
Age (year)	57	69	56	71
Gender	male	female	male	male
Location of hepatic lobe	right	left	right	NM
Tumor size (cm)	2.5	3.9	2.5	4.5
US				
Echogenicity	slight hyperechoic	NM	Hypo- to isoechoic	NM
Tumor margin	unclear	unclear	irregular and unclear	NM
Homogeneity	heterogeneous	heterogeneous	homogeneous	NM
CEUS				
Contrast agent	Sonazoid	Sonazoid	Sonazoid	SonoVue
AP	hypervascularity	hypervascularity	hypervascularity	hypervascularity
PP	hypovascularity	isovascularity	hypervascularity	isovascularity
PVP	hypoechogenicity	hypoechogenicity	hypoechogenicity	hypoechogenicity
Enhancement in re-injection	Yes	No	Yes	/
Dynamic enhancement patterns	peripheral enhancement + centripetal filling	peripheral enhancement + centripetal filling	peripheral enhancement	peripheral enhancement

^1^ HSH—hepatic sclerosed hemangioma; US—ultrasound; CEUS—contrast-enhanced ultrasound; AP—arterial phase; PP—portal phase; PVP—post-vascular phase; NM—not mentioned.

**Table 2 diagnostics-12-02818-t002:** Review of clinical features for HSH ^1^.

Publishing Year	First Author	Patient Source	Patient Age (Year) ^2^	Tumor Size (cm) ^2^	Lesions/Patients ^3^	Male/Female ^3^	Cirrhosis ^3^	HBV/HCV/Others/None ^3^	Elevated AFP ^3^	Elevated CA19-9 ^3^	Elevated Transaminase ^3^	Elevated Bilirubin ^3^	Left/Right Lobe (Location) ^3^
2021	Jia C [14]	China	50.5 (17–63)	3.9 (1–36.5)	12/12	8/4	3	4/0/8/0	0	3	NM	NM	NM
2015	Miyamoto S [15]	Japan	76	5.9	1/1	1/0	0	0/0/0/1	0	0	0	0	0/1
2018	Miyata T [6]	Japan	60 (34–79)	2.7 (0.6–14.8)	5/5	4/1	0	2/0/0/3	0	0	NM	NM	0/5
2021	Kim YY [16]	Korea	62.1 (53–71.2)	2.0 (1.7–2.5)	18/18	9/9	3	5/0/3/10	NM	NM	NM	NM	NM
2017	Shimada S [17]	Japan	27	2.2	1/1	1/0	0	0/0/1/0	0	0	1	1	0/1
2019	Toumi N [18]	Tunisia	17	80	1/1	0/1	NM	NM	0	0	0	0	1/0
2014	Ridge C.A. [19]	Ireland	65 (41–78)	3.2 (1.2–10.3)	12/12	2/10	NM	NM	NM	NM	NM	NM	NM
2010	Hida H [20]	Japan	75	3	1/1	0/1	0	NM	0	0	NM	NM	0/1
2018	Navale P [5]	USA	69	3.2	1/1	0/1	0	NM	0	1	NM	NM	0/1
2018	Sugo H [4]	Japan	39	1.7	1/1	0/1	0	0/0/0/1	0	0	NM	0	0/1
2013	Shimada Y [21]	Japan	63	1.5	1/1	1/0	0	0/0/0/1	0	0	0	0	0/1
2018	Yugawa K [22]	Japan	48	1.3 (0.9–6.7)	3/1	1/0	NM	NM	0	0	1	0	2/1
2019	Hwang JA [1]	Korea	57.5 (56–73)	2 (1.2–2.5)	9/9	5/4	3	4/0/1/4	NM	NM	NM	NM	NM
2018	Ozaki K [7]	Japan	57	4.5	1/1	1/0	0	0/0/0/1	0	0	0	0	NM
2010	Jin SY [23]	Korea	52	3.8	1/1	1/0	NM	1/0/0/0	1	0	0	0	NM
2019	Koyama R [3]	Japan	68	2.2	1/1	1/0	NM	NM	0	0	1	0	0/1
2015	Wakasugi M [24]	Japan	67	2.0 (1.1–2.8)	2/1	0/1	NM	NM	NM	0	NM	NM	1/1
2019	Renzulli M [25]	Italy	68	2.0	1/1	1/0	NM	NM	NM	NM	NM	0	0/1
2013	Song JS [26]	Korea	63	9.0	1/1	0/1	NM	0/0/0/1	0	NM	1	0	1/0
2011	Shin YM [27]	Korea	50	10.0	1/1	1/0	NM	0/0/0/1	0	NM	0	0	0/1
2012	Yamada S [2]	Japan	75	1.1	1/1	1/0	NM	1/0/0/0	0	0	0	NM	0/1
2015	Andeen N.K. [28]	America	60	3.5	1/1	0/1	NM	NM	NM	0	0	0	1/0
Total ^4^	/	/	58.8 (17–79)	3.9 (0.9–80)	76/73	38/35	9	17/0/13/23	1	4	4	1	6/17

^1^ HSH—hepatic sclerosed hemangioma; NM—not mentioned; HBV—hepatitis B virus; HCV—hepatitis C virus; AFP—alpha-fetoprotein; CA19-9—carbohydrate antigen 19-9. ^2^ For the patient with multi-lesions, the age and size indicate mean value and range. ^3^ The values below the index (column) represent the number of cases that meet the condition of that index. ^4^ Some data are not available in some of the literature. Therefore, for columns with NM present, the total number is not the exact total number of all cases. Only the total number for which the indicator was given was calculated (exclude “not mentioned” cases).

**Table 3 diagnostics-12-02818-t003:** Review of CT/MR imaging features for HSH ^1^.

Publishing Year	First Author	Lesions ^2^	Heterogeneous/Homogeneous ^2^	“Clear Margin”/“Irregular or Unclear” ^2^	Calcification ^2^	Capsular Retraction ^2^	AP Enhancement in CT/MRI^2^	Hypointense/Non-Hypointense in T1-Weighted Image ^2^	Hyperintense/Hypointense in T2-Weighted Image ^2^	Hypointense/Isointense in HBP Image ^2^	Using only CT/only MRI/Both CT and MRI ^2^	Centripetal/Peripheral Enhancement Patterns ^2^
2021	Jia C	12	5/7	3/9	3	1	12	NM	8/0	NM	4/3/5	3/NM
2015	Miyamoto S	1	1/0	0/1	NM	NM	1	1/0	1/0	1/0	0/0/1	NM/1
2018	Miyata T	5	NM	0/3	NM	3	4	5/0	5/0	5/0	0/0/5	3/4
2021	Kim YY	18	NM	4/14	NM	NM	18	16/2	9/9	15/3	0/18/0	12/6
2017	Shimada S	1	1/0	0/1	NM	NM	1	NM	NM	1/0	0/0/1	NM/1
2019	Toumi N	1	0/1	1/1	1	NM	1	1/0	0/1	NM	0/1/0	1/1
2014	Ridge C.A.	12	NM	NM	NM	NM	7	12/0	5/6 ^3^	NM	0/12/0	4/9
2010	Hida H	1	1/0	1/0	NM	NM	1	1/0	1/0	NM	0/0/1	1/1
2018	Navale P	1	1/0	NM	1	NM	1	1/0	1/0	NM	0/0/1	NM
2018	Sugo H	1	1/0	1/0	NM	NM	1	1/0	1/0	1/0	0/0/1	0/1
2013	Shimada Y	1	NM	0/1	NM	NM	1	1/0	1/0	1/0	0/0/1	NM/1
2018	Yugawa K	3	3/0	0/3	NM	NM	3	3/0	3/0	3/0	0/0/3	0/3
2019	Hwang JA	9	3/0	2/7	NM	3	8	8/1	6/3	9/0	0/9/0	3/5
2018	Ozaki K	1	1/0	0/1	NM	NM	1	1/0	1/0	1/0	0/0/1	0/1
2010	Jin SY	1	NM	NM	NM	NM	1	1/0	1/0	1/0	0/0/1	NM
2019	Koyama R	1	NM	0/1	NM	0	1	1/0	1/0	1/0	0/0/1	0/1
2015	Wakasugi M	2	2/0	0/2	NM	NM	2	2/0	2/0	NM	0/0/2	0/2
2019	Renzulli M	1	1/0	1/0	NM	NM	1	1/0	1/0	1/0	0/1/0	0/1
2013	Song JS	1	1/0	0/1	0	NM	1	/	/	/	1/0/0	1/1
2011	Shin YM	1	1/0	0/1	0	NM	1	1/0	1/0	NM	0/0/1	1/1
2012	Yamada S	1	NM	NM	NM	NM	1	1/0	1/0	1/0	0/0/1	0/1
2015	Andeen N.K.	1	NM	NM	NM	1	1	1/0	1/0	NM	0/0/1	1/1
Total ^4^	/	76	22/8	13/46	5	8	69	59/3	50/19	41/3	5/44/27	30/42

^1^ HSH—hepatic sclerosed hemangioma; NM—not mentioned; HBP—hepatobiliary phase; CT—computed tomography; MRI—magnetic resonance imaging; AP—arterial phase. ^2^ For the patient with multiple lesions, the age and size indicate mean value and range. ^3^ Of the 12 patients with HSH lesions, one patient did not have available FSE T2W imaging; therefore, the total number is 11. ^4^ Some data are not available in some of the literature. Therefore, for columns with NM present, the total number is not the exact total number of all cases. Only the total number for which the indicator was given was calculated (exclude “not mentioned” cases).

## Data Availability

The data presented in this study are available from the corresponding author upon request. The data are not publicly available due to privacy.

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
