# Peer review of "Added Value of Ultrasound-Based Multimodal Imaging to Diagnose Hepatic Sclerosed Hemangioma before Biopsy and Resection"

_diagnostics, 2022, doi:10.3390/diagnostics12112818_

Round 1

Reviewer 1 Report

In my opinion It should be better to obtain the written informer consent from the patients. 

Author Response

Thank you very much for the reviewer’s kind remind. We have got the written informed consent from the patient who we detailed described using ultrasound elasticity, CEUS, and B-flow diagnosing HSH. Please see the signed informed consent as follows. We also uploaded it as a supplementary material.

We also changed "oral informed consent" to "written informed consent" in revised manuscript (line 73). 

Reviewer 2 Report

More quantitative work should be performed to reach sound conclusion.

Author Response

Thank you very much for the reviewer’s constructive comments. We admit our quantitative work was insufficient and lack of statistical results (We wrote this point as a limitation in our manuscript as“We only carried out descriptive analysis, rather than statistical analysis, on the imaging features of HSH” Lines 319-320). The actual causes of this defect are as follows.

Firstly, HSH is a rare disease. To the best of our knowledge, there is no published  literature on the diagnosis of HSH with ultrasound elasticity and B-flow imaging. There are only three published HSH cases using CEUS as diagnostic modality. Therefore, it is difficult to carry out a reliable comparative statistical analysis between our case and other studies. Second, so far, our patient is the first to be diagnosed with HSH using “ultrasound-based multimodal imaging”. Additional examinations (such as tumour biomarkers, PET-CT) would certainly yield more valuable information and attract readers. However, the premise of any research is to fully consideration of the interests of patients rather than publishing articles, such as taking additional unnecessary examinations and unnecessary cost to patient. Based on this consideration, we have not collected a lot of data of this case. To enrich data as a compensation, we were fortunately to have the patient’s consent for close follow-up for two years (and long-term follow-up is continuing).

Reviewer 3 Report

The authors report the interesting clinical case of a quite uncommon hepatic benign lesion and they provide a review of the literature on the topic. The strengh of the article is the author's perspective on imaging which reflects experience with advanced ultrasound-based techniques which are commonly disregarded in so many clinical contexts. The main drawback of the work is poor standardization of results due to the case-report nature and the limited panoramic view and reproducibility of the ultrasound-based imaging modalities.

I congratulate with the authors for a well-written article. I have the following suggestions:

1 - Title

Multimodal imaging technologies can be well used to diagnose hepatic sclerosed hemangioma before biopsy and resection

To be more consistent with reported data I would suggest to change with:

Added value of ultrasound-based multimodal imaging to diagnose hepatic sclerosed hemangioma before biopsy and resection

2 - abstract

I aknowledge the large author's experience with CEUS. If on the one hand Japanese peaple are particularly suited for ultrasound imaging thanks to physical features, on the other hand the authors should recognize that Ultrasound present many technical limitation in the general population due to poor acoustic window, steatosis and overweight. The readers should be aware that a milestone in non-invasive liver imaging is represented by Computed Tomography which is not free from iodizing radiation exposure.

I would suggest to change the first sentence of the abstract as follows:

Imaging methods have the overwhelming advantages of being non-invasive in the diagnosis of hepatic lesions and, thanks to technical developments in the field of Ultrasounds, also radiation exposure can be avoided in many clinical situations.

3 - abstract conclusion

Please remove terms such as "we hope" and rephrase only using sentences consistent with the reported data.

4 - time selection for literature review

I think this is the most critical point since the selection of 2018-2022 period cuts a significant body of literature based on the same technological tools. I would start at least from 2010.

5 - paragraph titles

paragraph 3.2 and 3.3 titles are misspelled: "using CEUS diagnose or using CT/MR diagnose" are nonsense. Please correct

6 - The described clinical entity is very uncommon and, especially in the high risk population of oncologic or chronic liver disease patients, I would always suggest to perform a percutaneous needle biopsy for lesion characterization in the presence of pattern of malignancy (wash in and wash out). The features provided by the authors (microvascular features, centripetal filling on CEUS) could be more useful in the low-risk population to safely address patients to follow up. I would highlight this comment in discussion.

7 - In the discussion section the authors should remark the higher temporal resolution, and even spatial resolution (with high-frequency transducers), of Ultrasound and CEUS compared to other panoramic techniques. Ultrasound is able to assess lesion perfusion almost by a microvascular perspective.

8 - At the same time the authors should remark the limitation of Ultrasound in difficult abdomens. 

9 - Minor English review for grammar and spellings check

Author Response

Thank you for your insight and valuable comments. Please see attachment of our response and revised manusript.

Reviewer 4 Report

This is a descriptive study of a case of HSH and the imaging features in particular with CEUS. The authors have also compared this to two other published cases with CEUS performed. In addition, the authors have reviewed 10 papers covering 52 lesions in 47 patients where imaging with CT/MRI was available.

As a case report for a single case, the authors have done extensive work to document the imaging and pathological features, and on its own would be suitable for publication as a case report of a rare clinical condition. They have also added the descriptive features on US elastography.

The comparison with the other two published cases where CEUS was described is useful. However, with conflicting features described in the three cases, it would have been better for the authors to have also compared the CT and MRI features for these 3 cases so that a more comprehensive observation can be made, rather than to place these features in Table 2 and 3 in comparison with all the other published papers which confuses the observations. This is especially relevant since the authors are proposing that CEUS in conjunction with CT/MRI can improve diagnostic accuracy.

The results are confusing in the way it is presented. For Figure 1 it will be easier for the reader to appreciate the features if these are separated into different figures for the different imaging modalities rather than to put 9 images into one figure.

For Table 1, it well describes the features on CEUS amongst three patients but would also be useful if the CT/MRI features of these 3 patients specifically were also included.

The discussion section is poorly written and difficult to understand. Some of the conclusions made are not substantiated by the observations. 

Overall there is a need for improvement in writing style and grammar.

Author Response

We are much appreciate for your constructive and professional comments. We learned much. We uploaded our response and revised manuscript in attachment.

Round 2

Reviewer 1 Report

no more comment 

Reviewer 4 Report

thank you for making the revisions. i understand your limitations